# Cascade Testing for Fragile X Syndrome in a Rural Setting in Cameroon (Sub-Saharan Africa)

**DOI:** 10.3390/genes11020136

**Published:** 2020-01-28

**Authors:** Karen Kengne Kamga, Séraphin Nguefack, Khuthala Minka, Edmond Wonkam Tingang, Alina Esterhuizen, Syntia Nchangwi Munung, Jantina De Vries, Ambroise Wonkam

**Affiliations:** 1Division of Human Genetics, Department of Pathology & Institute of Infectious Disease and Molecular Medicine (IDM), Faculty of Health Sciences, University of Cape Town, Cape Town, Rondebosch 7701, South Africa; KRNKEN003@myuct.ac.za (K.K.K.); mnikakhuthala@gmail.com (K.M.); wonkamedmond@yahoo.fr (E.W.T.); alina.esterhuizen@uct.ac.za (A.E.); nchangwisyntia@yahoo.com (S.N.M.); 2Department of Paediatrics, Faculty of Medicine and Biomedical Sciences, University of Yaoundé 1, Yaoundé 1937, Cameroon; seraphin_nguefack@yahoo.fr; 3National Health Laboratory Service, Groote Schuur Hospital, Cape Town, Observatory 7925, South Africa; 4Department of Medicine, Faculty of Health Sciences, University of Cape Town, Cape Town, Rondebosch 7701, South Africa; jantina.devries@uct.ac.za

**Keywords:** fragile X syndrome, genetic counselling, full mutation, premutation, Cameroon, Africa

## Abstract

Fragile X Syndrome (FXS), an X-linked dominant monogenic condition, is the main genetic cause of intellectual disability (ID) and autism spectrum disorder (ASD). FXS is associated with an expansion of CGG repeat sequence in the Fragile X Mental Retardation gene 1 (*FMR1*) on chromosome X. Following a neuropediatric assessment of two male siblings who presented with signs of FXS that was confirmed with molecular testing, we provided cascade counselling and testing to the extended family. A total of 46 individuals were tested for FXS; among them, 58.70% (*n* = 27) were females. The mean age was 9.4 (±5) years for children and 45.9 (±15.9) years for adults. Pedigree analysis suggested that the founder of these families was likely a normal transmitting male. Four out of 19 males with clinical ID were confirmed to have a full mutation for FXS, while 14/27 females had a pathologic CGG expansion (>56 CGG repeats) on one of their X chromosomes. Two women with premature menopause were confirmed of being carriers of premutation (91 and 101 CGG repeats). We also identified maternal alleles (91 and 126 CGG repeats) which expanded to a full mutation in their offspring (>200 CGG repeats). This study is a rare report on FXS from Africa and illustrates the case scenario of implementing genetic medicine for a neurogenetic condition in a rural setting.

## 1. Introduction

Fragile X Syndrome (FXS) is the most common monogenic X-linked condition which causes variable degrees of Intellectual Disability (ID), autism spectrum disorder (ASD), delay in acquisition of speech and other cognitive skills; affecting one in 7143 males and one in 11,111 females [1,2,3,4,5]. Some individuals with FXS may present characteristics of facial appearance such as, a large forehead and prominent ears. Moreover, some of these signs may only occur after puberty which is the case of adolescent males who may develop macro-orchidism. Hence, the diagnosis of FXS cannot rely exclusively on clinical features but will depend on molecular investigations for confirmation [6,7]. 

Fragile X Syndrome is associated with Cytosine-Guanine-Guanine (CGG) repeat sequence expansion located in the 5′ untranslated region (UTR) in the Fragile X Mental Retardation gene 1 (*FMR1*), at the Xq27 position [8,9]. This expansion which accounts for over 98% of all FXS cases exist in four allelic forms base on the length of the CGG repeats: normal (5–44), intermediate (45–55), pre-mutation (56–200), and full mutation (>200) [10,11,12]. Individuals with a full mutation may have methylated FMR*1* genes which turn off the production of mRNA that will translate to protein. The lack of the gene product, Fragile X Mental Retardation Protein (FMRP), is responsible for the clinical features [13,14].

Fragile X Syndrome follows the traditional pattern of X-linked inheritance. Males who inherit the mutation will be affected, and they will transmit the mutation to all their daughters. Also, females who carry the mutation may present with some degree of ID or behavioral problems and will transmit this mutation to half of their children [9,15]. However, the risk of expansion of the CGG repeats in a premutation allele to a full mutation overlays the transmission pattern of this syndrome [8,15]. Research on FXS in African population is limited probably due to scarce genomic and genetic services on the continent [16,17,18], and few qualified personnel [19]; only a few scholars have identified families living with FXS through clinical research [20,21,22]. It is likely that with the developing medical genetic services in countries such as Cameroon [18], this situation will gradually improve.

Cameroon is a Central African country which spans almost equally in two main geographical zones; the equatorial rainforest in the south and the tropical savanna and the Sahel region in the north. Its population was estimated to be 25,216,237 in 2018 [23]. Also known as “Africa in miniature”, Cameroon has a diverse cultural and linguistic heritage which mimics the heterogeneity found in Africa [24]. The health-care system in Cameroon is organized into the public, private and traditional sectors without universal health insurance coverage. Hence, patients depend on financial support and caregiving from family members and regularly consult traditional healers [25]. Poverty in Cameroon affects more than 50% of the rural population and up to 30% of the urban population, which implies that the necessary medical care for patients may not be satisfied due to the endured financial burden [25,26]. Besides communicable diseases like malaria, HIV-AIDS and TB, Cameroon, like many other developing countries are facing a transition with a growing burden of chronic non-communicable diseases, some of which are of genetic origin [18]. Yet studies show a poor knowledge of genetic diseases and genetic tests among medical students and physicians in Cameroon [27].

In Cameroon, the pediatric neurology unit at the Yaoundé gynaeco-obstetric and pediatric hospital has a longstanding interest in children with developmental delay [22]. Since 2011, this service has been following up two male siblings, who subsequently had a positive molecular diagnosis for FXS. While investigating these children’s condition, we realized that in their extended family, a “Royal family” from Western Cameroon, there were several individuals with similar FXS clinical presentation. Interestingly, the founder of the family happened to be the Chief of the village who had 25 wives and was the maternal great grandfather of the two affected boys. It is a believed in the village that the high number of FXS is a curse that was placed on the princesses by the Chief, because they refused to morn one of his servants who was intellectually disabled.

As part of an ancillary care of a social science research project initially designed to understand the community’s knowledge of FXS in that particular setting, and as a result of a pressing demand from the family members, we provided genetic counselling and testing for extended relatives in this large family [28]. Cascade testing, which is a systematic process of identifying individuals at risk of contracting a hereditable condition [29], has successfully been practiced in programs aimed at identifying individuals susceptible of having a genetic condition [30,31,32]. In this paper we aimed to describe the results of this cascade testing for FXS in a rural setting in Cameroon, and to discuss implications of genetic research for rare conditions in Africa.

## 2. Material and Methods

### 2.1. Ethical Approvals

The study was performed in accordance with the Declaration of Helsinki. Ethical approval for the study was obtained from the Institutional Committee for Health Research (no. 698/CIERSH/DM/2018) in Yaoundé, Cameroon; and the University of Cape Town’s Faculty of Health Sciences’ Human Research Ethics Committee (HREC: 782/2017). Written informed consent was obtained from all participants who were 21 years of age or older, and from parents or guardians in cases of minors, with verbal assent from participants, including permission to publish photographs.

### 2.2. Participants

Scoping meetings aiming at exploring approaches to involve the extended family members in the cascade counselling and testing for FXS was held in 2018 in Yaoundé, between the mother of the boys initially identified to have FXS, and the first, second and last authors, who are respectively general practitioner, neuropediatrician, and medical geneticist. At her discretion she decided to inform family members of the diagnosis. Some family members expressed the interest to be tested, and the above medical professionals were invited to one of their family reunions for formal introduction. A pamphlet explaining the process was provided to family members, that explains FXS and the genetic testing procedure. Family members who expressed interest to be tested had a clinical consultation at the local government hospital. Each participant had a pre-counselling with the objective to provide information about FXS and to identify possible family and community support. Demographic information was collected via the use of a structured questionnaire, that included sociodemographic variables (age, sex, religion, profession, highest level of education), anthropometric variables (weight, height), and relevant clinical variables. In particular, the participant’s neurologic status was evaluated by determining the levels of support they needed to attain an optimal personal functioning [33,34]. We classified ID as normal, mild, moderate, severe or profound (Appendix A). Figure 1 illustrates the flow diagram for the recruitment of participants. Family pedigree was drawn using the software Cyrillic 3.0.400, Ken Lange, Los Angeles, CA, USA (Figure 2).

All tested participants later received a written report stating whether they were normal, had a premutation or a full mutation for FXS, during a post genetic counselling session. The counselling and feedback of results were performed by the General Practitioner, and as appropriate, further care was done by the neuropediatrician.

### 2.3. Molecular Analysis

Peripheral blood samples were collected from consenting participants between August 2018 and August 2019. Samples were sent to the Division of Human Genetics, University of Cape Town, Cape Town, South Africa, where DNA was extracted from leucocytes following standardized protocols (according to chemagic™ 360 Nucleic Acid Extractor, Waltham, MA, USA). Sequencing and determination of CGG repeats were performed in an accredited molecular diagnosis laboratory; the National Health Laboratory services (NHLS), at the Groote Schuur Hospital, in Cape Town, South Africa. Analysis of the disease associated CGG *FMR1* repeat region was performed using the AmplideX *FMR1* PCR kit (Asuragen, Inc, Austin, TX, USA) and capillary electrophoresis (ABI3500 Genetic Analyser, ThermoFisher Scientific, Maltham, USA). Fragment analysis and sizing were performed using GeneMapper® Software 5 (ThermoFisher Scientific, Maltham, NY, USA). Alleles were categorized as normal, intermediate, premutation or fully expanded according published repeat size ranges [35,36].

### 2.4. Statistical Analysis 

Descriptive statistics was used, with Epi-info 7.2, CDC, USA.

## 3. Results

### 3.1. Participants Sociodemographic

Table 1 describes the participants. A total of 46 participants were included in the study, 23 children (<18 years) and 23 adults. Of these, 19 were male and 27 were females (sex ratio of 0.7). The mean age was 9.4 (±5) years for children and 45.9 (±15.9) years for adults.

### 3.2. Molecular and Pedigree Analysis 

Of the 46 participants who underwent Fragile X carrier testing, 28 (60.87%) were normal (CGG repeats < 55); among them, the CGG repeat ranged from 5 to 55 in normal *FMR1* alleles, with the most prevalent alleles being 29 repeats (19.6%), followed by 30 repeats (13.04%) and 31 repeats (13.04%) (Figure 3).

Four of the 19 males (21.1%) presented with a full mutation (CGG repeats >200) and were all classified as having a severe ID; one of them had a family with 3 female children, suggesting a certain social acceptance and tolerance of disability in this specific setting. 

Among the female participants 10/27 and 4/27 females had a premutation and a full mutation respectively, giving a carrier frequency for FXS of 51.8% among females. Nine of the 14 females who either had a premutation or a full mutation were normal (64.3%), while the rest (*n* = 5/14) had a mild ID. Of the 5 females with mild ID, 3 had a full mutation for FXS (>200 CGG repeat), and the other 2 had a premutation (91, 101 CGG repeats) and presented with premature menopause. No case of Fragile X-associated Tremor Ataxia Syndrome (FXTAS) was reported in the families.

With these results, we were able to describe the pattern of maternal allele transmission. There was an increase in the length of the mutated maternal allele from one generation to the other (Table 2). The pedigree in conjunction with the molecular analysis also support that the great grandfather of our proband, who was the leader of a traditional community in Cameroon, with 25 wives, must have been a Normal Transmitting male (Figure 2). Indeed, seven of his daughters were screened for the *FMR1* gene mutation in the present study. Six out of 7 were carriers of a pre-mutation (58, 81, 91, 100, 101, 110, CGG repeats) while one had a 200 CGG repeats.

## 4. Discussion 

This study is a rare attempt to delineate FXS in an extended family in Africa, and particularly illustrates a case of implementation of genetic counselling and genetic medicine in rural setting where molecular diagnosis for a genetic disease is inaccessible, via an intra-African collaboration between pediatricians and medical genetics team from Cameroon and South Africa respectively. The identification of a proband with a careful analysis of the pedigree led to the diagnosis of premutation carriers in 10 females. These individuals were previously not aware of their FXS status. By knowing their diagnosis, female carriers were able to understand some relevant clinical presentations such as the risk for premature ovarian failure, and possible risk of ID in their offspring. Although most children with the premutation do not have neurodevelopmental deficits, recent studies have suggested that some children will manifest learning problems, shyness or anxiety [37]. This information is important to guide reproductive options such as seeking children earlier in life and to seek for genetic diagnosis before birth, which is possible in Cameroon [18].

The CGG repeat distribution varies among different population. In this Cameroonian family, the most prevalent allele in normal members of this family was 29 repeats. Hung et al. [38] also reported a high prevalence of 29 CGG repeat in a population of Chinese women. Moreover, we could identify two families with premutation that was above 70 repeats and expanded to a full mutation in one generation. Our finding is concurrent with previous reports that an increased number of CGG repeats give an increased instability of alleles from one generation to the next, resulting in alleles with an increased number of repeats in the progenies [35,38]. This information is important in counselling carriers since they will need this information to guestimate their chances of having a child with a full mutation.

However, pre-testing and post-testing genetic counselling of FXS can be challenging, specifically in counselling of the variable phenotypes associated with FXS, especially in female patients with Premutation or Full mutation [39]. In our sample size, 14 females had pathologic CGG expansions on their x chromosomes. Five out of fourteen had a mild form of ID. The clinically normal female who had a full mutation could be attributed to an unbiased X chromosome inactivation [40,41]. A concern with counselling of premutation carriers is explaining the symptoms of Fragile X-associated Premature Ovarian Insufficiency (FXPOI) and Fragile X-associated Tremor Ataxia Syndrome (FXTAS). FXTAS mostly affects males in their fifties while women still present with premature ovarian insufficiency (POI) by the age of 40 [42,43].

### 4.1. Practice Implications 

The identification of a positive case of FXS has initiated cascade testing in a rural population where molecular diagnosis is inaccessible to the population. Information provided in the present study are important for reproductive decision in individual tested for FXS. The ability to detect a young female at risk of developing POI before the age of child baring, increases their options of planning to have a family through premarital screening and prenatal diagnosis. Furthermore, through counselling, several relatives were happy to take the *FMR1* mutation test and inform other family members who were skeptical about the screening. This study urges the need for screening program for targeted FXS which can be extended to other inheritable conditions. This can also lead to the development and strengthening of other genetic services like genetic counselling, prenatal diagnosis and neonatal screening in Africa.

### 4.2. Research Recommendation 

Further studies detailing the traditional vs. modern molecular knowledge of FXS specifically with respect to the “curse” story need to be properly explored in this setting using qualitative research and ethnological research approach. In addition, psychosocial burden and possible stigma associated to FXS in this family will also require a specific investigation. Moreover, lived experience after the return of FXS results to participants in the present study will need to be formally evaluated. Additionally, given that FXS is a rare disease and we expect to have few patients, we advocate for regional collaborations in order to form a pool of patients with FXS and other rare genetic conditions, for future studies. This large family could help in exploring potential genetic modifiers of FXS, including differential methylation status, X inactivation and Adenine-Guanine-Guanine (AGG) interruptions in female carriers.

### 4.3. Study Limitations 

One of the study limitations was the refusal of some affected family members to be tested. So, the population presented in this paper represent only a small fraction of the affected individuals in this family. Besides, we did not probe for Adenine-Guanine-Guanine (AGG) interruptions and the methylation statues in our samples which could have aided in predicting the risk of full mutation expansion from premutation and the phenotypic presentation of the different individuals, and possibly explain the relatively high proportion of females with mild to moderate mental retardation.

## 5. Conclusions 

This study describes the pattern of genetic transmission of FXS in an exceptionally large Cameroon family and the proof of concept of a successful cascade genetic counselling, and molecular testing in a rural setting in Africa. The study is a case scenario of implementing genetic medicine for a neurogenic condition in a rural setting in Africa. Moreover, findings from this research will help increase our understanding of challenges associated with genetic counselling, public knowledge of genetics, and return of genetic results, as well as the psychosocial burden of rare genetic disease in an African setting.

## Figures and Tables

**Figure 1 genes-11-00136-f001:**
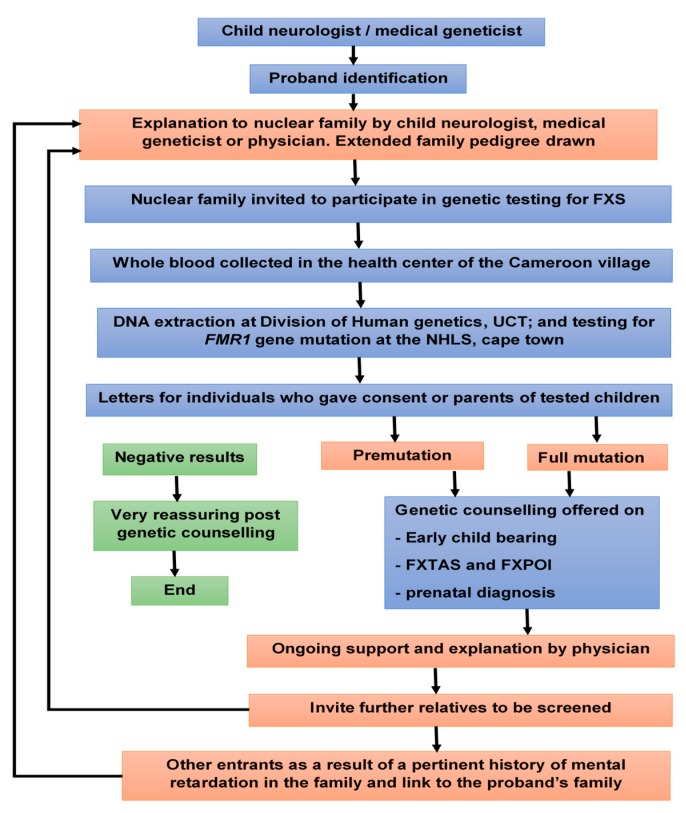
Flow chart of cascade testing for Fragile X Syndrome in Cameroon.

**Figure 2 genes-11-00136-f002:**
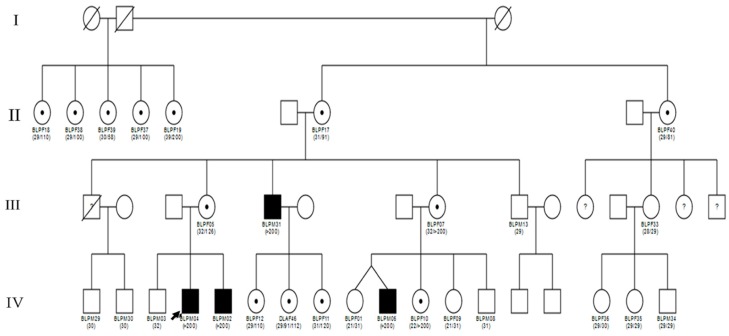
Description of our proband’s family. The number in brackets below the symbols as the number of CGG repeats in the *FMR1* gene.

**Figure 3 genes-11-00136-f003:**
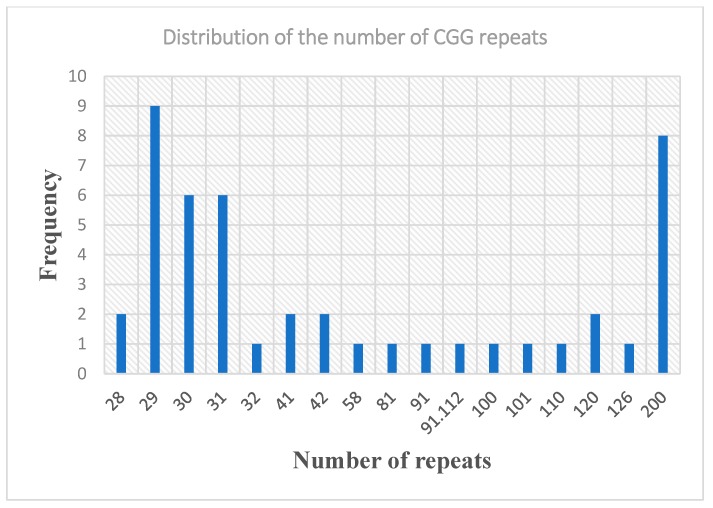
Distribution of Cytocine-Guanine-Guanine (CGG) repeat in the extended family.

**Table 1 genes-11-00136-t001:** Description of the study participants: FXS mutation, and Intellectual Disability status.

Variable		Mutation Pattern			Total
		N	PM	FM	
**Subject**	Male	15	0	4	19
	Female	13	10	4	27
**ID Status of Males**	Absent	14	0	0	14
	Mild ID	1	0	0	1
	Severe ID	0	0	4	4
**ID Status of Females**	Absent	13	8	1	22
	Mild ID	0	2	3	5
	Severe ID	0	0	0	0

ID: Intellectual Disability, FM: Full Mutation, PM: Pre-Mutation, N: Normal.

**Table 2 genes-11-00136-t002:** Consequences of transmitting a maternal PM allele.

Maternal Allele (CGG Repeat) Transmitted	Child Allele (CGG rRpeat)		CGG Repeat Increase	Maternal Allele (CGG Repeats) Not Transmitted
	Male	Female		
91 (I)		126	+35	31
	>200		+>109	31
		>200	+>109	31
126 (II)	>200		+>74	32
	>200		+>74	32
>200 (III)	>200		0	32
		>200	0	32

CGG: Cytocine-Guanine-Guanine; PM: Premutation.

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
