# Peer review of "Cascade Testing for Fragile X Syndrome in a Rural Setting in Cameroon (Sub-Saharan Africa)"

_genes, 2020, doi:10.3390/genes11020136_

Round 1
Reviewer 1 Report
The authors present a single case of fragile X cascade screening of an extended family in Cameroon. Results are consistent with expectations in this disease and no novelty at the molecular level is presented. According to the authros the novelty of their work is that it is one of the first of such studies in Africa. Thus, the interest of the paper would be more at health management and sociological level.
Suggestions:
Molecular and genetic analysis should be extended to the determination of AGG repeat interruptions. The authors claim that the CGG repeat analysis was performed using the Assuragen kit. Information of the AGG interruptions can be easily extracted from this kit and it is unclear why this was not done.
The perceived main interest of this work is the sociological and cultural context in which the study was performed. Further detail of this context would be of great interest.
Author Response
|
Comments reviewer 1 |
Response to reviewer 1 |
|
This is a very interesting paper of a Royal pedigree from Cameroon Africa. As the authors mentioned there is very little information in the literature of African pedigrees so this is of great interest. There are a few English phrases that need correction as follows: On line 42 the word "can" should be changed to "will" because all daughters of carrier males are carriers. |
Thanks for the positive comments.
The requested change on Line 42 has been corrected, accordingly. |
|
So, this points out that one daughter of the carrier maternal great grandfather has a full mutation which is not possible. They should recheck the DNA on this woman as to the exact number of CGG repeats. If it is 200 then this is believable but not higher. Another possibility is that there is none paternity or perhaps the princess that he laid a curse on was really the mother of the child with the full but then this woman with the full should have another premutation allele from her father. Anyway, this parentage of this daughter with the full needs further investigation. |
Thank you for this insightful comment. We can confirm that one daughter of the great grandfather of our proband had 200 CGG repeats. Unfortunately, we were not able to count CGG repeats above 200. We plan to follow this in subsequent pedigree analysis and molecular studies, as our laboratory techniques improve. |
|
Other correction: line 160: normal should be inserted before members of this family. |
The change requested has been made. Thanks! |
|
page 2 line 47 scares should be changed to" fears" of genomic.... |
Apology for this typo. We meant limited services. We have corrected the spelling from scares to scarce (line 47) |
|
line 132 FXTAS should be written out as fragile X-associated tremor ataxia |
FXTAS has been written as suggested. …. Fragile X-associated Tremor Ataxia Syndrome
|
Reviewer 2 Report
This is a very interesting paper of a Royal pedigree from Camaroon Africa. As the authors mentioned there is very little information in the literature of African pedigrees so this is of great interest. There are a few English phrases that need correction as follows: On line 42 the word "can" should be changed to "will" because all daughters of carrier males are carriers. So this points out that one daughter of the carrier maternal great grandfather has a full mutation which is not possible. They should recheck the DNA on this woman as to the exact number of CGG repeats. If it is 200 then this is believable but not higher. Another possibility is that there is non paternity or perhaps the princess that he laid a curse on was really the mother of the child with the full but then this woman with the full should have another premutation allele from her father. Anyway this parentage of this daughter with the full needs further investigation.
Other correction: line 160: normal should be inserted before members of this family.
page 2 line 47 scares should be changed to" fears" of genomic....
line 132 FXTAS should be written out as fragile X-associated tremor ataxia
Overall this is a very interesting paper that should be puclished.
Author Response
|
Comments reviewer 2 |
Response reviewer 2 |
|
The authors present a single case of fragile X cascade screening of an extended family in Cameroon. Results are consistent with expectations in this disease and no novelty at the molecular level is presented. According to the authors the novelty of their work is that it is one of the first of such studies in Africa. Thus, the interest of the paper would be more at health management and sociological level. |
Thanks for this positive comment. |
|
Suggestions: Molecular and genetic analysis should be extended to the determination of AGG repeat interruptions. The authors claim that the CGG repeat analysis was performed using the Assuragen kit. Information of the AGG interruptions can be easily extracted from this kit and it is unclear why this was not done. |
Thanks for this suggestion. The aim of the study was to obtain the number of CGG repeats which will help diagnose Fragile X Syndrome in this royal family. Regarding the presence/absence of the AGG raw data, whilst this can be seen in the electrophoretic traces, it is not used for actual result interpretation in the diagnostic setting, and not included in the patients reports. However, we plan to follow up with AGG interruptions identification in future molecular studies in this family. |
|
The perceived main interest of this work is the sociological and cultural context in which the study was performed. Further detail of this context would be of great interest. |
Thanks for the suggestion, we have now added the following background information in the introduction: Cameroon is a Central African country which spans almost equally in two main geographical zones; the equatorial rain forest in the south and the tropical savanna and the Sahel region in the north. Its population was estimated to be 25,216,237 in 2018 [23]. Also known as "Africa in miniature", Cameroon has a diverse cultural and linguistic heritage which mimics the heterogeneity found in Africa [24]. The health-care system in Cameroon is organized into the public, private and traditional sectors without universal health insurance coverage. Hence, patients depend on financial support and caregiving from family members and regularly consult traditional healers [25]. Poverty in Cameroon affects more than 50% of the rural population and up to 30% of the urban population, which implies that the necessary medical care for patients may not be satisfied due to the endured financial burden [25, 26]. Besides communicable diseases like malaria, HIV-AIDS and TB, Cameroon, like many other developing countries are facing a transition with a growing burden of chronic non-communicable diseases, some of which are of genetic origin [18]. Yet studies show a poor knowledge of genetic diseases and genetic tests among medical students and physicians in Cameroon [27].
The references were also updated accordingly.
|
|
Reviewer 2 suggests that manuscript should undergo extensive |
As requested by the reviewers, the manuscript was proofread by two colleagues whose native language is English. They corrected the grammar and reformulated a few sentences. The changes are highlighted in red in the updated manuscript that we are summiting. |
Round 2
Reviewer 1 Report
Introduction has been improved with a perspective of the country and its health system. English language has been improved. Suggested AGG repeat interruption determination has not been performed.